# COVID-19 pandemic in Saint Petersburg, Russia: Combining population-based serological study and surveillance data

Anton Barchuk[1,7]*, Dmitriy Skougarevskiy[1], Alexei Kouprianov[2], Daniil Shirokov[3], Olga Dudkina[1], Rustam Tursun-zade[1], Mariia Sergeeva[4], Varvara Tychkova[4], Andrey Komissarov[4], Alena Zheltukhina[4], Dmitry Lioznov[4], Artur Isaev[5,6], Ekaterina Pomerantseva[5], Svetlana Zhikrivetskaya[5], Yana Sofronova[5], Konstantin Blagodatskikh[5], Kirill Titaev[1], Lubov Barabanova[3], Daria Danilenko[4]

1 European University at St. Petersburg, St. Petersburg, Russia, 2 Independent Researcher, St. Petersburg, Russia, 3 Clinic "Scandinavia" (LLC Ava-Peter), St. Petersburg, Russia, 4 Smorodintsev Research Institute of Influenza, St. Petersburg, Russia, 5 Center of Genetics and Reproductive Medicine GENETICO LLC, Moscow, Russia, 6 Human Stem Cells Institute, Moscow, Russia, 7 Petrov National Research Medical Center of Oncology, Pesochny, St. Petersburg, Russia

* abarchuk@eu.spb.ru

**Data Availability Statement:** study data and code is available online https://github.com/eusporg/spb_covid_study20.

## Abstract

### Background

The COVID-19 pandemic in Russia has already resulted in 500,000 excess deaths, with more than 5.6 million cases registered officially by July 2021. Surveillance based on case reporting has become the core pandemic monitoring method in the country and globally. However, population-based seroprevalence studies may provide an unbiased estimate of the actual disease spread and, in combination with multiple surveillance tools, help to define the pandemic course. This study summarises results from four consecutive serological surveys conducted between May 2020 and April 2021 at St. Petersburg, Russia and combines them with other SARS-CoV-2 surveillance data.

### Methods

We conducted four serological surveys of two random samples (May–June, July–August, October–December 2020, and February–April 2021) from adults residing in St. Petersburg recruited with the random digit dialing (RDD), accompanied by a telephone interview to collect information on both individuals who accepted and declined the invitation for testing and account for non-response. We have used enzyme-linked immunosorbent assay Corona-Pass total antibodies test (Genetico, Moscow, Russia) to report seroprevalence. We corrected the estimates for non-response using the bivariate probit model and also accounted the test performance characteristics, obtained from independent assay evaluation. In addition, we have summarised the official registered cases statistics, the number of hospitalised patients, the number of COVID-19 deaths, excess deaths, tests performed, data from the ongoing SARS-CoV-2 variants of concern (VOC) surveillance, the vaccination uptake, and

**Funding:** Polymetal International plc funded the serological study. The main funder had no role in study design, data collection, data analysis, data interpretation, writing of the report or decision to submit the publication. The European University at St. Petersburg, clinic "Scandinavia", Smorodintsev Research Institute of Influenza and Genetico had access to the study data. The European University at St. Petersburg had final responsibility for the decision to submit for publication. Part of this study performed at Smorodintsev Research Institute of Influenza was funded by the Russian Ministry of Science and Higher Education as part of the World-class Research Center program: Advanced Digital Technologies (contract No. 075152020904, dated 16.11.2020). The study's funder had no role in study design, data collection, data analysis, data interpretation, or writing of the report.

**Competing interests:** Anton Barchuk reports personal fees from AstraZeneca, MSD, and Biocad outside the submitted work. Artur Isaev, Ekaterina Pomerantseva and Svetlana Zhikrivetskaya report a pending patent for the test system (ELISA) for detecting antibodies specific to the SARS-COV-2 in a biological sample. Other authors have no conflict of interest to declare. This does not alter our adherence to PLOS One policies on sharing data and materials. It also complies with the manuscript submission guidelines of PLOS One.

St. Petersburg search and mobility trends. The infection fatality ratios (IFR) have been calculated using the Bayesian evidence synthesis model.

## Findings

After calling 113,017 random mobile phones we have reached 14,118 individuals who responded to computer-assisted telephone interviewing (CATI) and 2,413 provided blood samples at least once through the seroprevalence study. The adjusted seroprevalence in May–June, 2020 was 9.7% (95%: 7.7–11.7), 13.3% (95% 9.9–16.6) in July–August, 2020, 22.9% (95%: 20.3–25.5) in October–December, 2021 and 43.9% (95%: 39.7–48.0) in February–April, 2021. History of any symptoms, history of COVID-19 tests, and non-smoking status were significant predictors for higher seroprevalence. Most individuals remained seropositive with a maximum 10 months follow-up. 92.7% (95% CI 87.9–95.7) of participants who have reported at least one vaccine dose were seropositive. Hospitalisation and COVID-19 death statistics and search terms trends reflected the pandemic course better than the official case count, especially during the spring 2020. SARS-CoV-2 circulation showed rather low genetic SARS-CoV-2 lineages diversity that increased in the spring 2021. Local VOC (AT.1) was spreading till April 2021, but B.1.617.2 substituted all other lineages by June 2021. The IFR based on the excess deaths was equal to 1.04 (95% CI 0.80–1.31) for the adult population and 0.86% (95% CI 0.66–1.08) for the entire population.

## Conclusion

Approximately one year after the COVID-19 pandemic about 45% of St. Petersburg, Russia residents contracted the SARS-CoV-2 infection. Combined with vaccination uptake of about 10% it was enough to slow the pandemic at the present level of all mitigation measures until the Delta VOC started to spread. Combination of several surveillance tools provides a comprehensive pandemic picture.

## Introduction

The COVID-19 pandemic in Russia has already resulted in 500,000 excess deaths [1], with more than 5.6 million cases registered officially by July 2021 [2]. Surveillance based on case reporting has become the core method for monitoring the pandemic in Russia and globally. However, the actual spread of SARS-CoV-2 is challenging to measure as case definitions, testing strategies, and capacity are not comparable between countries and in the different periods [3]. Population-based studies using representative samples of the population combined with serological assessment for the presence of SARS-CoV-2 may provide an unbiased estimate of the actual disease spread and help estimate the true disease burden as well as the infection fatality rate (IFR) [4–7]. Unfortunately, national serological studies to assess the prevalence of SARS-CoV-2 in Russia were not yet published. Given a considerable territory, it is not likely that the pandemic develops similarly across the country. Therefore, different studies are needed to explore seroprevalence and the pandemic course in big cities and less densely populated regions. Saint Petersburg is the second-largest city in Russia, with the first SARS-CoV-2 case registered on March 5, 2020. Seroprevalence study conducted in St. Petersburg between May 27 and June 26, 2020, estimated that not more than 10% of the population had contracted

the SARS-CoV-2 [8]. These findings were in line with seroprevalence estimates in other European studies summarised in the systematic review [9], which revealed only 82 studies of higher quality out of 404 studies included in the meta-analyses. Lack of sample representativeness and methods to correct participants' characteristics and test performance limited the quality for most assessed studies.

Official case count and serological studies are not the only methods for SARS-CoV-2 surveillance. Cause-specific COVID-19 mortality is another valuable statistic to assess the pandemic impact. However, it may be biased in different healthcare settings, especially when definitions for COVID-19 death are not comparable. Using excess mortality, i.e. quantifying deaths from all causes relative to a recent historical benchmark, can help avoid this bias [1, 10]. St. Petersburg was one of the two Russian regions with the most reliable reporting of COVID-19 mortality [11]. SARS-CoV-2 variants of concern (VOC) monitoring is another critical surveillance tool that turned to become crucial in the later stages of the COVID-19 pandemic when new, more transmissive VOCs started to spread rapidly [12, 13].

Novel auxiliary surveillance methods like search term trends to monitor the COVID-19 pandemic and mobility trends to monitor the effects of mitigation measures and population behaviour can also be helpful [14, 15]. For example, search terms and mobility trends are available for St. Petersburg. However, these low-barrier research methods are often criticised for the lack of validity [16].

This study summarises the four consecutive rounds of population-based serological study based on two representative samples of adults residing in St. Petersburg, Russia, between May 2020 and April 2021. In addition, we combine the seroprevalence estimates with all other available surveillance data: official case count, hospitalisation data, SARS-CoV-2 VOCs monitoring data, COVID-19 specific mortality, excess mortality, vaccination uptake, mobility trends, and search term trends. Thus, we aim to assess whether different surveillance tools gave consistent insights for the course of the epidemics in the fourth largest European city with more than 5 million residents.

## Materials and methods

### Seroprevalence of anti-SARS-CoV-2 antibodies

St. Petersburg serological study settings and design are described in detail in our previous report [8]. In brief, St. Petersburg COVID-19 study is population-based epidemiological survey of a random sample from the adult population to assess the seroprevalence of anti-SARS-CoV-2 antibodies. The study was based on a phone-based survey and an individual invitation to the clinic for blood sample collection. Eligible individuals were adults residing in St. Petersburg older than 18 years and recruited using the random digit dialling (RDD) method. RDD was accompanied by the computer-assisted telephone interviewing (CATI) to collect the information on both individuals who accepted and declined the invitation for testing. Blood samples from the same population group were collected between May 25, 2020, and June 28, 2020, in the first cross-section "May–June 2020 survey" henceforth) and between July 20, 2020, and August 8, 2020, in the second "July–August 2020 survey" cross-section). Considering the risks of low response in the next planned cross-section, we created a new population sample applying the similar strategy of RDD followed by CATI ("October–December 2020 survey"). The initial response to the RDD was higher in autumn and winter 2020–2021 compared to the first cross-section in summer 2020. The fourth cross-section ("February–April 2021 survey") involved individuals from both population samples invited between February 15, 2021, and April 4, 2021. Repeated blood sampling allowed seroconversion assessment for individuals who tested positive in previous surveys. Also, in this cross-section, some

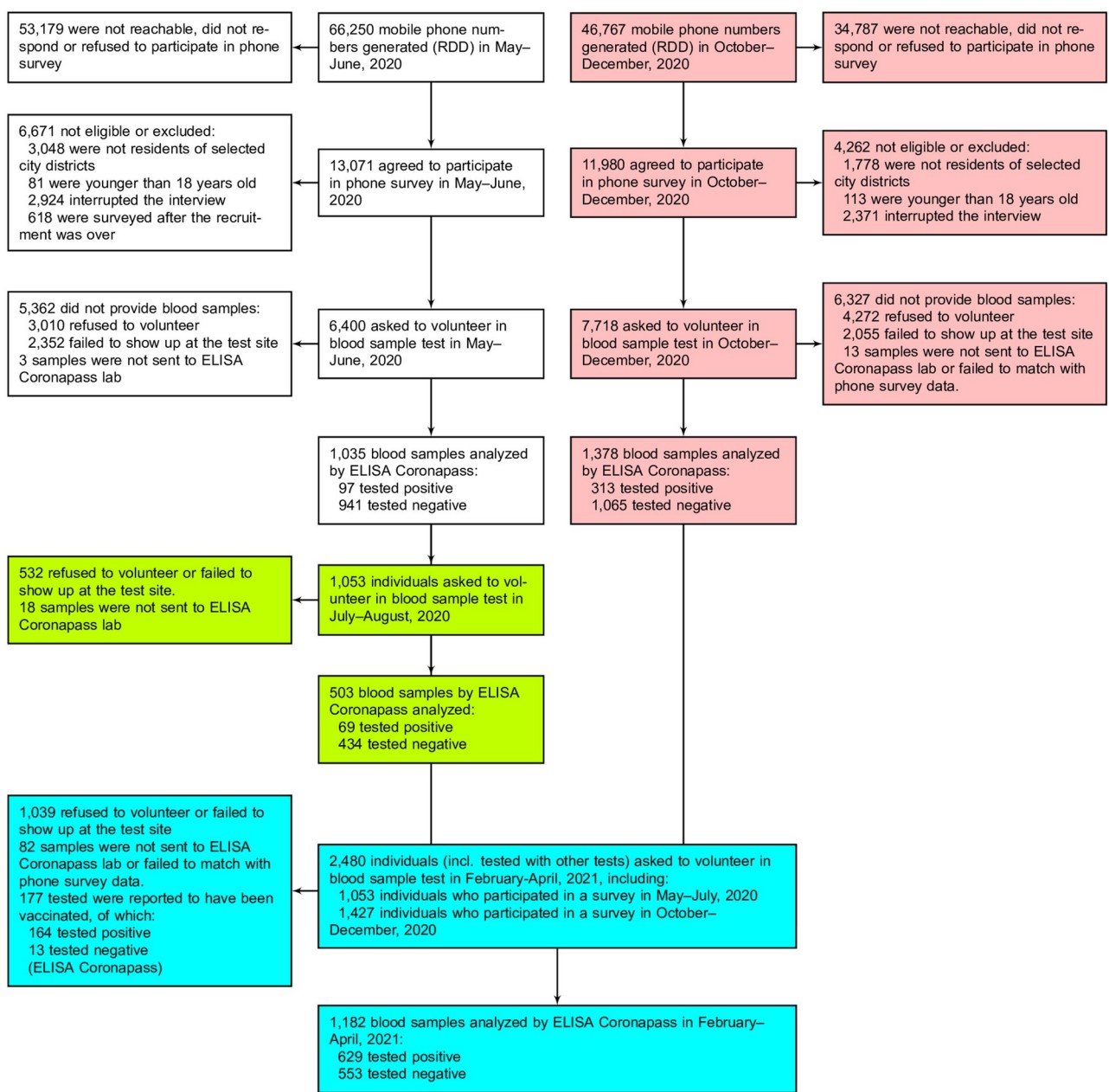

**Fig 1. Flow chart of participants' progress through the St. Petersburg seroprevalence study (color codes study cross-sections: White—cross-section 1 (2020-05-25—2020-06-28), green—cross-section 2 (2020-07-20—2020-08-08), pink—cross-section 3 (2020-10-12—2020-12-06), blue—cross-section 4 (2020-02-15—2020-04-04).).**

participants reported at least one vaccine shot. They were included in the study as non-responders as the initial survey does not fully address the characteristics associated with vaccination status. However, the vaccinated individuals were still tested. The participant flow for all four cross-sections is reported in Fig 1. The full study protocol is available online (https://eusp.org/sites/default/files/inline-files/EU_SG-Russian-Covid-Serosurvey-Protocol-CDRU-001_en.pdf).

## Laboratory tests

During the four surveys, we assessed anti-SARS-CoV-2 antibodies using three different assays. Even though our report was selected among studies of higher quality in the recent systematic review, a significant limitation was related to the absence of own test performance validation [9]. We conducted a validation that revealed the decrease in sensitivity for one of the assays [17]. Finally, to report seroprevalence, we selected enzyme-linked immunosorbent assay (ELISA Coronapass) CoronaPass total antibodies test (Genetico, Moscow, Russia) that detects total antibodies (the cutoff for positivity 1.0) and is based on the recombinant SARS-CoV-2 spike protein receptor binding domain (Department of Microbiology, Icahn School of Medicine at Mount Sinai, New York, NY, USA). We used ELISA Coronapass through all four surveys. We also used the results of our validation study to correct the seroprevalence estimate for test performance. Sensitivity is equal to 92% and specificity to 100% for ELISA Coronapass (for full validation see [17]).

## Surveillance data related to SARS-CoV-2 monitoring

We summarised the data that included the official registered cases statistics, the number of patients hospitalised, the number of COVID-19 deaths, excess deaths, and tests performed for COVID-19 detection. Although this information was not available from one source, we used a combination of different sources to restore the pandemic course in St. Petersburg. We have also used the leading Russian search engine Yandex search history in St. Petersburg region to obtain search trends for three terms: "loss of smell", "smell", and "saturation". In addition, Yandex provided mobility trends for St. Petersburg from the open data from Yandex, Apple, and Otomono (https://yandex.ru/company/researches/2020/cities-activity). Finally, we obtained data from the ongoing the Smorodintsev Research Institute of Influenza SARS-CoV-2 VOCs surveillance in St. Petersburg [18, 19]. Data sources are described in detail in our Supplementary material.

## Infection fatality ratios

We used the information on the official COVID-19 mortality and derived excess mortality to estimate the IFR. IFR was calculated for the four periods covered by our seroprevalence surveys. We treated the true number of deaths as an interval censored random variable bound downwards/upwards by the number of deaths 14 days after the cross-section start/end date (see S1 Appendix).

## Statistical analysis

The sample size calculations and statistical analysis plan for the serological survey were described in detail in our previous report [8]. The initial sample size of 1550 participants was calculated assuming the hypothetical prevalence of 20% to obtain the resulting sampling error of 2% using a 95% confidence interval. The actual sample size was lower, which resulted in a maximum error of about 4% when hypothetical seroprevalence reached 40%. The study's primary aim was to assess the seroprevalence based on antibody tests accounting for non-response bias and test sensitivity and specificity. Non-response was evaluated by comparing answers provided during the CATI by those who visited the test site and all surveyed individuals, estimated using a binomial probit regression of individual agreement to participate in the study and offer their blood sample on their observable characteristics. In the first report, we described the variables that we had chosen to estimate the correction. The observable characteristics associated with response and positivity were reported any disease symptoms before

the test and the COVID-19 testing history. We used similar variables to correct the seroprevalence estimates for non-response during all four cross-sections. To account possible sample non-representativeness in a sensitivity analysis, we computed raking weights to match the survey age group and educational attainment proportions in the 2016 representative survey of the adult city population with R package `anesrake` used to compute the weights. The original report also explored individual risk factors for test positivity in the sample participants who completed clinic paper-based surveys. This report assessed individual risk factors using a binomial probit regression used to estimate seroprevalence. Standard errors were computed with the delta method. For IFR computations we relied on a Bayesian evidence synthesis model [20] described in S1 Appendix.

## Ethical considerations

The Research Planning Board approved the study of the European University at St. Petersburg (on May 20, 2020) and the Ethics Committee of the Clinic "Scandinavia" (on May 26, 2020). All research was performed following the relevant guidelines and regulations. Eligible individuals were adults (older than 18 years). Written informed consent was obtained from all participants of the seroprevalence study. The study was registered with the following identifiers: Clinicaltrials.gov (NCT04406038, submitted on May 26, 2020, date of registration—May 28, 2020) and ISRCTN registry (ISRCTN11060415, submitted on May 26, 2020, date of registration—May 28, 2020). Official statistics, VOCs monitoring data, search terms trends, and mobility trends were obtained from open sources as aggregated data. Analysis based on open-source aggregated data does not require additional ethical permission in Russia.

## Data sharing

All analyses were conducted in R, study data and code is available online (https://github.com/eusporg/spb_covid_study20).

## Results

### Seroprevalence of antibodies to SARS-CoV-2

The resulting 14,118 individuals responded to CATI questionnaire—6,400 in the first population sampling and 7,718 in the second (see S1 Appendix for details regarding missing records on variables of interest). The respondents represent city population in terms of their gender, employment status, and household size, but were younger than the adult city population as of 2016 and had higher levels of educational attainment (see S1 Appendix). Overall, 2,413 individuals provided blood samples through the seroprevalence study course that were analysed using ELISA Coronapass: 1,035 in the first May–June 2020 survey and 503 of them in the second July–August survey, and 1,378 newly recruited participants in the third October–December survey. Finally, samples from 1,182 participants from previous surveys were collected and analysed in February–April 2021.

The adjusted seroprevalence in May–June 2020 was 9.7% (95%: 7.7–11.7) and increased to 13.3% (95% 9.9–16.6) in July–August 2020. We noticed a major increase through the third (22.9% 95%: 20.3–25.5) and between the third and fourth cross-sections of the seroprevalence study (see Fig 3 and S1 Appendix for the weekly data), resulting in seroprevalence equal to 43.9% (95%: 39.7–48.0) in February–April 2021. Naïve antibodies seroprevalence to SARS-CoV-2 and seroprevalence corrected for non-response only and corrected for non-response and test performance are presented in Table 1.

**Table 1. Seroprevalence by study cross-section, ELISA Coronapass.**

| Serosurvey cross-section | Seroprevalence estimate | | | |
|---|---|---|---|---|
| | N interviewed / N tested | Naïve | Adjusted for non-response | Adjusted for non-response and test characteristics |
| (May 25, 2020—June 28, 2020) | 5951 / 988 | 10.6 (8.7–12.5) | 8.9 (7.1–10.8) | 9.7 (7.7–11.7) |
| 2 (July 20, 2020—August 8, 2020) | 5951 / 474 | 15.2 (12.0–18.4) | 12.2 (9.1–15.3) | 13.3 (9.9–16.6) |
| 3 (October 12, 2020—December 6, 2020) | 7110 / 1322 | 23.2 (20.9–25.5) | 21.0 (18.7–23.4) | 22.9 (20.3–25.5) |
| 4 (February 15, 2021—April 4, 2021) | 13412 / 1140 | 53.2 (50.3–56.1) | 40.4 (36.5–44.2) | 43.9 (39.7–48.0) |

Seroprevalence estimates adjusted through raking weights were similar and seroprevalence by different subgroups are available in S1 Appendix. History of any symptoms, history of COVID-19 tests, and non-smoking status were significant predictors for higher seroprevalence.

## Seroconversion results

The SARS-CoV-2 antibodies test results trajectories showed that most individuals remained seropositive with a maximum follow-up of 10 months (Fig 2). Among 177 participants who have reported at least one vaccine dose by the end of April, 2021, 92.7% (95% CI 87.9–95.7) were seropositive.

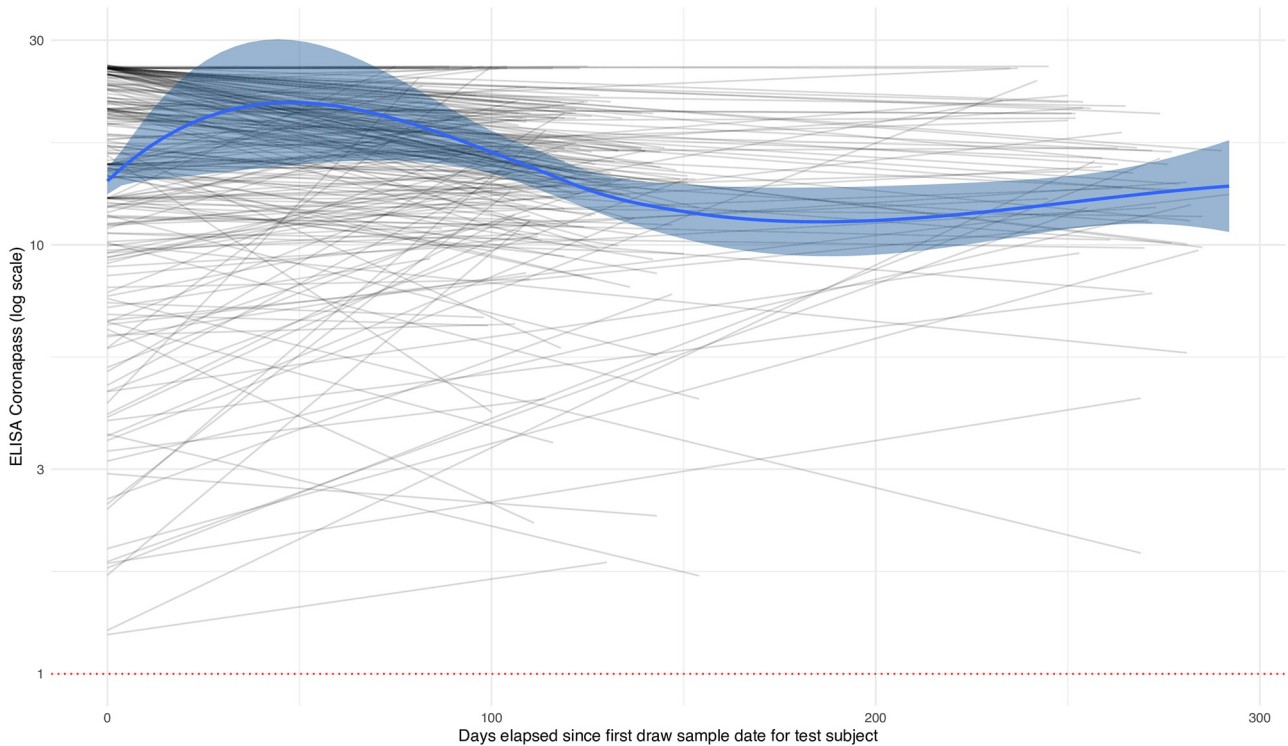

**Fig 2. Trajectories of antibodies to SARS-CoV-2 (ELISA Coronapass).** Grey lines are individual trajectories of study participants who tested positive at least once, excluding the 2020-07-20—2020-08-08 cross-section. Solid blue line is the loess smoother, blue areas report its 95% CI.

## Combining other sources of pandemic surveillance

The number of cases officially registered in the spring 2020 was much lower than in the autumn and the winter 2020–2021. Number of SARS-CoV-2 tests reached its maximum in the winter 2020–2021 in contrast to a relatively low number of tests reported in the spring 2020. Official case statistics contrast the number of hospitalisations, official deaths, and excess deaths reported in the spring 2020. The official number of cases, the number of hospitalisation and deaths from COVID-19 never reached zero between and after the pandemic waves. The number of COVID-19 deaths and excess deaths from all causes peaked in both periods and was in line with hospitalisation dynamics (Fig 3).

Internet-based search terms trends were in line with pandemic dynamics. They reflected the changes in hospitalisation and death count better than the official case count, especially during the spring wave (Fig 3). In addition, urban activity trends showed an apparent response to the first spring wave, somewhat less evident response during the second winter wave, and return to pre-pandemic activity levels in the late spring of 2021.

The SARS-CoV-2 circulating lineages diversity in 2020 was low. All samples from this period were attributed to the B.1 lineage and its sublineages. By autumn 2020 the number of PANGO lineages gradually increased with two Russian endemic—the B.1.397 and B.1.317. The Alpha VOC (B.1.1.7) was first detected in February 2021. The number of B.1.1.7 cases did not increase steeply but showed a gradual increase by April 2021. In February 2021, another lineage—AT.1, that has probably emerged in St. Petersburg was detected. The AT.1 was spreading rather quickly till April 2021, when B.1.617.2 (the Delta VOC) was first detected and substituted all other lineages by June 2021 (Fig 3).

## Infection fatality ratio

Using excess deaths data, the IFR was equal to 1.04 (95% CI 0.80–1.31) for the adult population for the whole pandemic period. IFR based on the official COVID-19 deaths counts was lower and amounted to 0.43% (95% CI 0.11–0.82). When we considered the entire population of the city rather than the adult population for IFR, we obtained the estimate of 0.86% (95% CI 0.66–1.08) based on the excess deaths data. Full results for IFR are reported in S1 Appendix. There was a clear upward trend in IFR by age. IFR was higher in men in all age groups.

## Discussion

Our study is the first comprehensive attempt to characterise the pandemic dynamics in the fourth largest European metropolitan area. We used all available sources for surveillance, including population-based seroprevalence study, the monitoring of SARS-CoV-2 VOCs, data on registered cases and deaths, relevant search term trends and city activity. Combining this data provides an overall global picture how the pandemic evolved through 2020 and 2021 in St. Petersburg. In April 2021, approximately one year after COVID-19, we estimated that about 45% contracted the SARS-CoV-2 infection in St. Petersburg, roughly 2.2 mln residents. Together with more than 10% vaccination uptake to that moment, less than 45% susceptibles were there in the population. Nevertheless, it was enough to avoid a new pandemic wave in the absence of mitigation measures till the spread of the Delta VOC (B.1.617.2) at the end of May 2021.

The first year of the COVID-19 pandemic in St. Petersburg can be characterised by two waves of similar intensity but different lengths. In the spring 2020, the first pandemic wave resulted in unprecedented mitigation measures and population reaction that helped flatten the pandemic curve and preserve the healthcare system functionality. As a result, the daily registered number of cases have plateaued in summer. It helped reorganise the hospital capacities

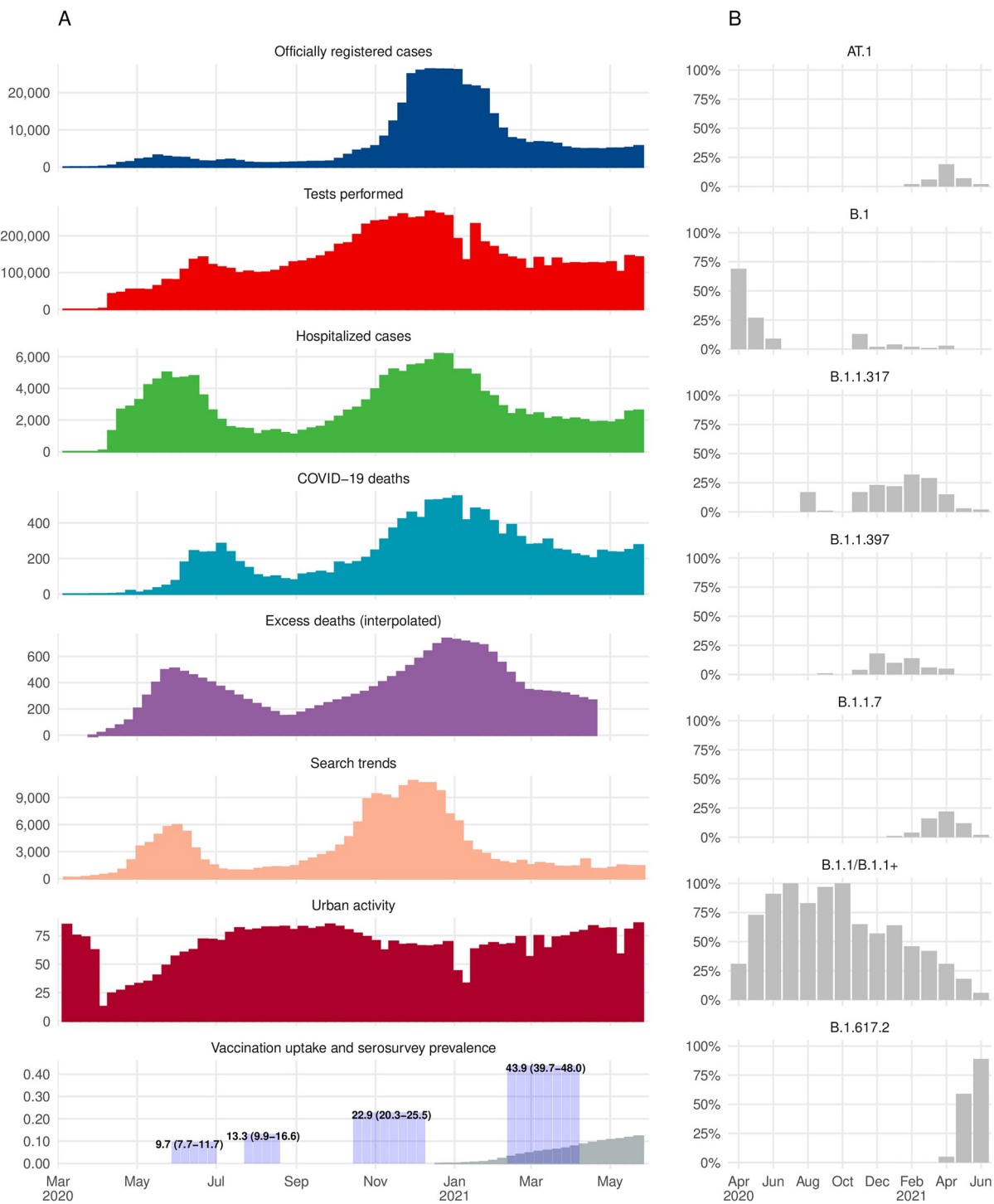

**Fig 3. Combining available surveillance data to monitor the pandemic course in St. Petersburg during March-May 2020–2021.** (A) Weekly data of officially registered cases, tests performed, hospitalised cases, COVID-19 deaths, interpolated excess deaths (from monthly data), search trends, urban activity, and vaccination uptake combined with seroprevalence estimates; (B) Monthly data on SARS-CoV-2 variants monitoring during April-June 2020–2021.

in St. Petersburg and prepare for the subsequent increase in the case count. The hospitals had to experience the entire load in autumn and winter, although many additional beds were allocated to COVID-19 patients. The number of daily cases plateaued during the winter holidays went down to 700–900 officially registered cases per day in the spring 2020. The halt of most mitigation measures has not resulted in the subsequent pandemic wave until the Delta VOC started to spread rapidly in May 2021. The new summer 2021 pandemic wave is yet to be analysed.

Possibly the number of individuals who already have antibodies to SARS-CoV-2, which was reported to be a strong protection marker from reinfection [21] combined with mitigation measures still in place in winter, has played its role in the pandemic dynamics in 2021 in St. Petersburg. In our study, we did not see any seroreversion events with a maximum follow-up of ten months, which is in line with some other studies [22]. Population-based vaccination was introduced in St. Petersburg in early 2021 and progressed slowly but involved primarily individuals who have not contracted the disease. Therefore, the sum of individuals seropositive after infection and the vaccinated individuals can approximate the number of protected individuals, yielding around 50–55% individuals with antibodies to SARS-CoV-2 by the end of April 2021. However, this approximation may not be valid in the future as more and more individuals who contracted the disease proceed to vaccination.

One of the surprising findings, which other studies reproduce [23, 24], is an association between seropositivity and smoking status. Seroprevalence was lower for smokers. That association was evident for both population samples in our study. Our study, however, does not answer the question, whether smokers are less likely to be infected or to develop less durable protection against infection [25], which is more likely given higher IFR in men who smoke more often in Russia. There are other characteristics associated with seroprevalence, but the nature of our cross-sectional study does not allow any causal conclusions.

Internet search term trends were quite reliably reflecting the pandemic's progress and predicted the increase in the number of hospitalisation and deaths for both waves in St. Petersburg. However, the Internet search term trends to monitor pandemics should be considered with caution [16]. This convenient surveillance option is compelling only in settings where the web-based search for medical conditions and symptoms is available and popular. Another critical limitation of the Internet search term trends lies in the spectrum of symptoms related to the disease of interest. For example, loss of smell is quite a distinct feature of SARS-CoV-2 infection. If the clinical manifestation of the infection caused by the new strains differ, surveillance strategies using search term trends should also change. It should also be acknowledged that internet search trends may predict the start of the epidemic wave well, but not its duration or overall burden.

More than 20,000 excess deaths have already been reported in St. Petersburg during the pandemic year [11]. The results of our seroprevalence study combined with the data on excess mortality give the IFR equal to 0.86% for the entire population, which is in line with other estimates across Europe [7, 24]. The IFR based on serological study results and excess mortality was stable for all four surveys. However, the official COVID-19 death count provided lower IFRs, which were not stable and was even lower during the pandemic waves. Thus, it seems that the number of deaths during the both waves was unprecedented for St. Petersburg to timely provide official data collection and cause of death specifications in mortality records.

We continue to monitor the pandemic in St. Petersburg using all available sources and plan to run the following survey to estimate the number of individuals with antibodies to SARS-CoV-2 after the summer wave. In addition, we aim to detect the herd immunity threshold in St. Petersburg if any exists given the Delta VOC basic reproductive number and diminished vaccine effectiveness [12].

Several possible limitations of our serological survey may require further explanation. Small sample size and high non-response rate compared to the number of phone numbers generated pose a challenge in two cases. First, when the obtained sample is small enough to make the study underpowered. Our sample size calculations show that under the 50% hypothetical prevalence scenario, our sampling error does not exceed 3% [8]. Second, a high non-response rate is a problem when there is an unaccounted selection on observables or unobservables into the tested subsample. Under our study design, we observe a rich set of characteristics of individuals to account for non-response. Our previous report rigorously addressed possible selection related to a low response rate in the serosurvey [8]. As a result, that report was selected among a few high-quality seroprevalence studies in the systematic review that addressed the quality of seroprevalence research [9].

In conclusion, our study provided an overall description of SARS-CoV-2 pandemic progression in the fourth largest European city—St. Petersburg, Russia, using all available surveillance sources, including a population-based serological study to assess the prevalence of antibodies to SARS-CoV-2. More than a half of the city's population had antibodies to the new coronavirus by April 2021, most of them due to prior infection. That was enough to control the SARS-CoV-2 at the present level of the mitigation measures only until the Delta VOC universal spread. When compared against the number of overall excess deaths, our seroprevalence estimates align with the IFR of 0.86%. Furthermore, the combination of different surveillance sources, including internet search term trends, provide a clear picture of the course of the SARS-CoV-2 pandemic in St. Petersburg.

## Supporting information

**S1 Appendix. Supplementary text, tables and figure.**
(PDF)

## Acknowledgments

We acknowledge personal support from Vitaly Nesis. We thank Alla Samoletova (European University at St. Petersburg) for administrative support and management of the study, Yulia Stepantsova (Chursina) for coordinating phone-based interviews, Lizaveta Dubovik, and Irina Shubina for the science communication. We thank the interviewers, nurses, general practitioners, and the Clinic "Scandinavia" personnel. We also thank all study participants.

## Author Contributions

**Conceptualization:** Anton Barchuk, Dmitriy Skougarevskiy, Daniil Shirokov, Yana Sofronova, Kirill Titaev, Lubov Barabanova, Daria Danilenko.

**Data curation:** Anton Barchuk, Dmitriy Skougarevskiy, Alexei Kouprianov, Daniil Shirokov, Mariia Sergeeva, Varvara Tychkova, Andrey Komissarov, Alena Zheltukhina, Dmitry Lioznov, Artur Isaev, Ekaterina Pomerantseva, Svetlana Zhikrivetskaya, Konstantin Blagodatskikh, Kirill Titaev.

**Formal analysis:** Anton Barchuk, Dmitriy Skougarevskiy, Alexei Kouprianov, Olga Dudkina, Rustam Tursun-zade, Andrey Komissarov, Dmitry Lioznov.

**Funding acquisition:** Anton Barchuk, Daniil Shirokov, Kirill Titaev, Lubov Barabanova, Daria Danilenko.

**Investigation:** Daniil Shirokov, Mariia Sergeeva, Varvara Tychkova, Alena Zheltukhina, Artur Isaev, Ekaterina Pomerantseva, Svetlana Zhikrivetskaya, Yana Sofronova, Konstantin Blagodatskikh.

**Methodology:** Anton Barchuk, Kirill Titaev, Lubov Barabanova, Daria Danilenko.

**Project administration:** Lubov Barabanova, Daria Danilenko.

**Resources:** Artur Isaev.

**Software:** Alexei Kouprianov, Olga Dudkina, Rustam Tursun-zade.

**Supervision:** Anton Barchuk, Dmitry Lioznov, Daria Danilenko.

**Validation:** Anton Barchuk, Dmitriy Skougarevskiy, Dmitry Lioznov, Daria Danilenko.

**Visualization:** Anton Barchuk, Dmitriy Skougarevskiy.

**Writing – original draft:** Anton Barchuk, Alexei Kouprianov, Olga Dudkina, Andrey Komissarov, Konstantin Blagodatskikh.

**Writing – review & editing:** Dmitriy Skougarevskiy, Daniil Shirokov, Rustam Tursun-zade, Mariia Sergeeva, Varvara Tychkova, Alena Zheltukhina, Dmitry Lioznov, Artur Isaev, Ekaterina Pomerantseva, Svetlana Zhikrivetskaya, Yana Sofronova, Kirill Titaev, Lubov Barabanova, Daria Danilenko.

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
