## [Decision Letter · Decision Letter 0]

7 Feb 2022

PONE-D-21-27726COVID­-19 pandemic in Saint Petersburg, Russia: combining population­-based serological study and surveillance dataPLOS ONE

Dear Dr. Barchuk,

Thank you for submitting your manuscript to PLOS ONE. After careful consideration, we feel that it has merit but does not fully meet PLOS ONE’s publication criteria as it currently stands. Therefore, we invite you to submit a revised version of the manuscript that addresses the points raised during the review process.

In particular, please address Reviewer 1's request to report the prevalence of different genotypes in Russia. Please also address all the comments by Reviewer 2, especially the one on the low participation rate for blood sample collection, and whether this is related to the high seroprevalence estimated for May-June 2020.

We look forward to receiving your revised manuscript.

Kind regards,

Siew Ann Cheong, Ph.D.

Academic Editor

PLOS ONE

Journal Requirements:

Polymetal International plc funded the serological study. The main funder had no role in study design, data collection, data analysis, data interpretation, writing of the report or decision to submit the publication. The European University at St. Petersburg, clinic "Scandinavia", Smorodintsev Research Institute of Influenza and Genetico had access to the study data. The European University at St. Petersburg had final responsibility for the decision to submit for publication. Part of this study performed at Smorodintsev Research Institute of Influenza was funded by the Russian Ministry of Science and Higher Education as part of the World­class Research Center program: Advanced Digital Technologies (contract No. 075152020904, dated 16.11.2020).

Polymetal International plc funded the serological study. The main funder had no role in study design, data collection, data analysis, data interpretation, writing of the report or decision to submit the publication. The European University at St. Petersburg,

clinic “Scandinavia”, Smorodintsev Research Institute of Influenza and Genetico had access to the study data. The European

University at St. Petersburg had final responsibility for the decision to submit for publication. Part of this study performed at

Smorodintsev Research Institute of Influenza was funded by the Russian Ministry of Science and Higher Education as part of

12

the World class Research Center program: Advanced Digital Technologies (contract No. 075152020904, dated 16.11.2020)

Polymetal International plc funded the serological study. The main funder had no role in study design, data collection, data analysis, data interpretation, writing of the report or decision to submit the publication. The European University at St. Petersburg, clinic "Scandinavia", Smorodintsev Research Institute of Influenza and Genetico had access to the study data. The European University at St. Petersburg had final responsibility for the decision to submit for publication. Part of this study performed at Smorodintsev Research Institute of Influenza was funded by the Russian Ministry of Science and Higher Education as part of the World­class Research Center program: Advanced Digital Technologies (contract No. 075152020904, dated 16.11.2020).

6. Thank you for stating the following in the Competing Interests section: 

Anton Barchuk reports personal fees from AstraZeneca, MSD, and Biocad outside the submitted work. Artur Isaev, Ekaterina Pomerantseva and Svetlana Zhikrivetskaya report a pending patent for the test system (ELISA) for detecting antibodies specific to the SARS-COV-2 in a biological sample. Other authors have no conflict of interest to declare. Other authors have no conflict of interest to declare.

Reviewers' comments:

Reviewer's Responses to Questions

**Comments to the Author**

1. Is the manuscript technically sound, and do the data support the conclusions?

Reviewer #1: Yes

Reviewer #2: Partly

2. Has the statistical analysis been performed appropriately and rigorously? 

Reviewer #1: Yes

Reviewer #2: I Don't Know

3. Have the authors made all data underlying the findings in their manuscript fully available?

Reviewer #1: Yes

Reviewer #2: Yes

4. Is the manuscript presented in an intelligible fashion and written in standard English?

Reviewer #1: Yes

Reviewer #2: No

5. Review Comments to the Author

Reviewer #1: This work on the prevalence of COVID-19 in Saint Petersburg is very interesting. The conclusions are probable; the authors conclude that before the epidemic episode, due to the Delta variant, 50% of the population was already infected. It would be interesting to have a table that refers to seroprevalences obtained in other countries because this prevalence seems to me to be extremely high, as well as the ISR in relation to the whole population. One of the interesting things that would probably be worth discussing is the role of hand washing at the beginning of the epidemic in the seroprevalence. The subjects who started to wash their hands more often at the beginning of the epidemic have a lower seroprevalence which is seen in the second study but which disappears in the third. It would have been important to know if there was here, as in other places (this is something I have observed) a decrease in the precautionary gestures of hand washing or the use of hydro-alcoholic solutions as the epidemic progressed. It would be interesting to report, based on the Gisaids analysis, the prevalence of the different genotypes in Russia, in general, and more specifically in Saint Petersburg, using the standardized relative incidence of SARS-CoV-02 variant data.

Reviewer #2: Thank you for the opportunity to revise this manuscript. Please find below comments and suggestions thay I hope may help improve the manuscript:

1) The manuscript is understandable, but the quality of the English can be improved (there are several typos and a few sentences are not correct).

2) The Authors state they wanted to "assess the different surveillance tools validity", but in fact, validity was not formally assessed or quantified. What was done was to see whether the different surveillance tools gave consistent insights for the course of the epidemics. So that sentence should be rephrased.

3) Figure 1: what does "(incl. tested with other tests)" mean? Is that explained anywhere in the text? The manuscript is quite long and dense, and I may have missed it, but I recommend explain it in detail.

4) The main limitation is the very low participation rate (over 112,000 invited, and only 1,182 had blood samples on all occasions), implying that the potential for selection bias is just huge. I understand that the Authors did their best to account for non-respondency (this is explained in the Methods and again mentioned in the Discussion), but when around 1% of invited people complete the study, it would be unfair to say that all is fine. This should be acknowledged more clearly in the Discussion, for instance by detailing how the results could be affected (e.g. if acceptance was linked to one suspect to having been infected because of risky contacts etc.).

5) The seroprevalence looks very high (especially on the earliest time points), and I suspect that the selection bias may have been played a major role. A seroprevalence equal to 9.7% in May-June 2020 is hardly credible. How does it compare, for instance, with data in Northern Italy, where COVID first started to circulate in Europe, or to data regarding healthcare personnel, who were highly exposed because of their job?

6) I doubt that 45% infected and 10% vaccinated (i.e. 55% immune) would be enough to establish herd immunity and stop circulation by itself. It is true that pre-Delta variants were not particularly contagious, but even with a reproductive number equal to 2, you would need over 65% immune to reach some herd immunity. I suppose some other factor was at play (mitigation measures, individual protection through masks, less survival of the virus because based on climatic factors, ...). Please elaborate on this.

7) With 43.9% seropositivity and the population of Saint Petersburg, and given around 1% of infection fatality ratio, I would expect more deaths than reported (and mentioned by the Authors), closer to 25.000 than 20.000.

8) Internet search trends look to predict well the start of the epidemic wave, but definitely not its duration and, therefore, the overall burden of it. This should be acknowledged.

6. PLOS authors have the option to publish the peer review history of their article (what does this mean?). If published, this will include your full peer review and any attached files.

Reviewer #1: No

Reviewer #2: No

---

## [Author Response · Author response to Decision Letter 0]

22 Mar 2022

PONE-D-21-27726 Response to Reviewers

COVID-19 pandemic in Saint Petersburg, Russia: combining surveillance and population-based serological study data in May 2020–April 2021

Anton Barchuk, Dmitriy Skougarevskiy, Alexei Kouprianov, Daniil Shirokov, Olga

Dudkina, Rustam Tursun-zade, Mariia Sergeeva, Varvara Tychkova, Andrey Komissarov,

Alena Zheltukhina, Dmitry Lioznov, Artur Isaev, Ekaterina Pomerantseva, Svetlana

Zhikrivetskaya, Yana Sofronova, Konstantin Blagodatskikh, Kirill Titaev, Lubov Barabanova,

and Daria Danilenko

Response to reviewers:

We would like to thank reviewers for the constructive and insightful comments on the manuscript. We have addressed all the raised issues and provided our itemized responses below.

Reviewer #1:

This work on the prevalence of COVID-19 in Saint Petersburg is very interesting. The conclusions are probable; the authors conclude that before the epidemic episode, due to the Delta variant, 50% of the population was already infected. It would be interesting to have a table that refers to seroprevalences obtained in other countries because this prevalence seems to me to be extremely high, as well as the ISR in relation to the whole population. 

Reply: We would like to point out that the seroprevalence between February 15, 2021 – April 4, 2021, was 43.9 (39.7–48.0) among the adult population of Saint-Petersburg. We decided to avoid systematic comparisons with other countries as the mitigation measures, the basic pattern of contacts, and vaccination rates were different. Therefore, they cannot be easily accounted for in such a comparison. The quality of seroprevalence reports was also heterogeneous, as suggested by the systematic review published in The Lancet Global Health (Chen et al., 2021). For example, the seroprevalence in Manaus, Brazil, by October 2020 was 76%, but it is likely to be biased upwards (Sabino et al., 2021, Lancet). At the same time, seroprevalence in Geneva, Switzerland on 1 June–7 July 2021 was about 66%, but only about 30% of study participants had antibodies of infection origin (Stringhini et al., 2021, Eurosurveillance). The control measures in St. Petersburg were enforced less strictly than in Geneva. At the same time, vaccination rates were lower, and the autumn 2020 winter 2021 wave caused more hospitalization and deaths in Petersburg than in Geneva.

Text added: “Seroprevalence after the first wave in St. Petersburg was similar to other European cities, e.g., Geneva, Switzerland. However, subsequent epidemic waves caused more hospitalization and deaths in St. Petersburg. Seroprevalence in Geneva, Switzerland, measured on 1 June–7 July 2021, was about 66%, but only about 30% of study participants had antibodies of infection origin.” 

One of the interesting things that would probably be worth discussing is the role of handwashing at the beginning of the epidemic in the seroprevalence. The subjects who started to wash their hands more often at the beginning of the epidemic have a lower seroprevalence which is seen in the second study but disappears in the third. It would have been important to know if there was here, as in other places (this is something I have observed) a decrease in the precautionary gestures of hand washing or the use of hydro-alcoholic solutions as the epidemic progressed. 

Reply: It is an excellent discussion point, but unfortunately, the nature of our study (cross-sectional) doesn’t allow any conclusion on the causal nature of handwashing and the risk of infection. We will add this point to our discussion part.

Text added: There are other characteristics associated with seroprevalence, but the nature of our cross­sectional study does not allow any causal conclusions.

It would be interesting to report, based on the Gisaids analysis, the prevalence of the different genotypes in Russia, in general, and more specifically in Saint Petersburg, using the standardized relative incidence of SARS-CoV-02 variant data.

Reply: Unfortunately, the information on different genotypes in Russia is limited due to a small number of sequences performed compared to other countries. However, It is worth mentioning that our report is authored by collaborators from Smorodintsev Research Institute of Influenza in St. Petersburg, Russia. They provided the majority of sequences reports from Russia to the Gisaid database, and we used this information in Figure 1B. For example, out of 408 B.1.1.7+Q sequences from Russia, 115 were from St. Petersburg. In addition, we have added a reference to the paper that describes the rise and spread of the SARS-CoV-2 AY.122 lineage in Russia (Klink et al. 2021). 

Reference added: Klink GV, Safina K, Nabieva E, Shvyrev N, Garushyants S, Alekseeva E, Komissarov AB, Danilenko DM, Pochtovyi AA, Divisenko EV, Vasilchenko LA. The rise and spread of the SARS-CoV-2 AY. 122 lineage in Russia. medRxiv. 2021 Dec 5.

Reviewer #2:

The manuscript is understandable, but the quality of the English can be improved (there are several typos and a few sentences that are not correct).

Reply: We went through the manuscript one more time to correct errors and typos as suggested by the reviewer. 

The Authors state they wanted to "assess the different surveillance tools validity", but in fact, validity was not formally assessed or quantified. What was done was to see whether the different surveillance tools gave consistent insights for the course of the epidemics. So that sentence should be rephrased.

Reply: We rephrased the sentence as suggested by the reviewer. 

Text added: Thus, we aim to assess whether different surveillance tools gave consistent insights for the course of the epidemics in the fourth largest European city with more than 5 million residents.

Figure 1: what does "(incl. tested with other tests)" mean? Is that explained anywhere in the text? The manuscript is quite long and dense, and I may have missed it, but I recommend explain it in detail.

Reply: By “incl. tested with other tests,” we meant that we included individuals who were tested using antibody tests other than the one we used to calculate seroprevalence (we used several antibody tests to ensure consistent and internally valid results). We agree that this information is confusing and the final number of participants is clear from the flowchart, so we deleted this text.

The main limitation is the very low participation rate (over 112,000 invited, and only 1,182 had blood samples on all occasions), implying that the potential for selection bias is just huge. I understand that the Authors did their best to account for non-responding (this is explained in the Methods and again mentioned in the Discussion), but when around 1% of invited people complete the study, it would be unfair to say that all is fine. This should be acknowledged more clearly in the Discussion, for instance by detailing how the results could be affected (e.g. if acceptance was linked to one suspect to having been infected because of risky contacts, etc.).

Reply: We would like to point out that about 15,000 individuals were invited to the blood test. 112,000 is the total number of mobile phone numbers generated using the RDD (random digit dial) electronic system, not the number of invitations. RDD is a reliable method for conducting sociological surveys. To ensure our survey sample is not biased, we compared it to the 2016 round of the comprehensive monitoring of living conditions household survey by Russia’s Federal State Statistics Service. 

The second step is an invitation to the blood sample, which around 2,300 participants accepted, and selection bias at this stage is inevitable. To understand the direction of non-response bias in our data, we estimated a binomial probit regression of individual agreement to participate in the study and offer their blood sample on observable characteristics. Our previous report described this study design in detail (Barchuk et al., 2021, Scientific Reports). Unfortunately, we did not have enough space to describe it in detail in this report. Nevertheless, some of the participant’s characteristics were linked both to an agreement to participate and to infection risk, and we took this into account in our study design and analysis. The analysis is described in detail in the supplementary material to our previous report (Barchuk et al., 2021, Scientific Reports). 

Text added: Our previous report rigorously addressed possible selection related to

a low response rate in the serosurvey]. As a result, that report was selected among a few high-­quality seroprevalence studies in the systematic review that addressed the quality of COVID-19 seroprevalence research [9]. 

The seroprevalence looks very high (especially on the earliest time points), and I suspect that the selection bias may have been played a major role. A seroprevalence equal to 9.7% in May-June 2020 is hardly credible. How does it compare, for instance, with data in Northern Italy, where COVID first started to circulate in Europe, or to data regarding healthcare personnel, who were highly exposed because of their job?

Reply: Seroprevalence equal to 9.7% in May-June 2020 in St. Petersburg is comparable to seroprevalence more than 10% in Madrid in April-May, 2020 (Pollan et al., 2020, Lancet) and to seroprevalence between 7 and 10% in Geneva, Switzerland in April-May 2020 (Stringhini et al., 2020, Lancet). 

Text added: “Seroprevalence after the first wave in St. Petersburg was similar to other European cities, e.g., Geneva, Switzerland. However, subsequent epidemic waves caused more hospitalization and deaths in St. Petersburg. Seroprevalence in Geneva, Switzerland, measured on 1 June–7 July 2021, was about 66%, but only about 30% of study participants had antibodies of infection origin.” 

I doubt that 45% infected and 10% vaccinated (i.e., 55% immune) would be enough to establish herd immunity and stop circulation by itself. Indeed, pre-Delta variants were not particularly contagious, but even with a reproductive number equal to 2, you would need over 65% immune to reach some herd immunity. I suppose some other factor was at play (mitigation measures, individual protection through masks, less survival of the virus based on climatic factors, ...). Please elaborate on this.

Reply: We agree that the statement about herd immunity is inaccurate. We need to mention that it’s the combination of population immunity and some level of mitigation measures, individual protection, and continued self-isolation in some social groups. 

Text added: “Approximately one year after the COVID-19 pandemic about 45\\% of St.~Petersburg, Russia residents contracted the SARS-CoV-2 infection. Combined with vaccination uptake of about 10\\% it was enough to slow the pandemic at the present level of the mitigation measures until the Delta VOC started to spread.”

“That was enough to control the SARS-CoV-2 {without} at the present level of the mitigation measures only until the Delta VOC universal spread.”

With 43.9% seropositivity and the population of Saint Petersburg, and given around 1% of infection fatality ratio, I would expect more deaths than reported (and mentioned by the Authors), closer to 25.000 than 20.000.

Reply: We infer IFR based on the number of deaths (excess mortality or official statistics) and individuals infected (our seroprevalence estimates). First, we would like to highlight that we are only using information from the adult population, and an IFR of 1% is related to the adult population. Therefore, it is 0.86% for all citizens, as mentioned in Table S2. 

Internet search trends look to predict well the start of the epidemic wave, but definitely not its duration and, therefore, the overall burden of it. This should be acknowledged.

Reply: We agree with this point and added this to the discussion section. 

Text added: It should also be acknowledged that internet search trends may predict the start of the epidemic wave well, but not its duration or overall burden.

References 

Barchuk A, Skougarevskiy D, Titaev K, Shirokov D, Raskina Y, Novkunkskaya A, Talantov P, Isaev A, Pomerantseva E, Zhikrivetskaya S, Barabanova L. Seroprevalence of SARS-CoV-2 antibodies in Saint Petersburg, Russia: a population-based study. Scientific reports. 2021 Jun 21;11(1):1-9.

Chen X, Chen Z, Azman AS, Deng X, Sun R, Zhao Z, Zheng N, Chen X, Lu W, Zhuang T, Yang J. Serological evidence of human infection with SARS-CoV-2: a systematic review and meta-analysis. The Lancet Global Health. 2021 May 1;9(5):e598-609.

Klink GV, Safina K, Nabieva E, Shvyrev N, Garushyants S, Alekseeva E, Komissarov AB, Danilenko DM, Pochtovyi AA, Divisenko EV, Vasilchenko LA. The rise and spread of the SARS-CoV-2 AY. 122 lineage in Russia. medRxiv. 2021 Dec 5.

Pollán M, Pérez-Gómez B, Pastor-Barriuso R, Oteo J, Hernán MA, Pérez-Olmeda M, Sanmartín JL, Fernández-García A, Cruz I, de Larrea NF, Molina M. Prevalence of SARS-CoV-2 in Spain (ENE-COVID): a nationwide, population-based seroepidemiological study. The Lancet. 2020 Aug 22;396(10250):535-44.

Sabino EC, Buss LF, Carvalho MP, Prete CA, Crispim MA, Fraiji NA, Pereira RH, Parag KV, da Silva Peixoto P, Kraemer MU, Oikawa MK. Resurgence of COVID-19 in Manaus, Brazil, despite high seroprevalence. The Lancet. 2021 Feb 6;397(10273):452-5.

Stringhini S, Wisniak A, Piumatti G, Azman AS, Lauer SA, Baysson H, De Ridder D, Petrovic D, Schrempft S, Marcus K, Yerly S. Seroprevalence of anti-SARS-CoV-2 IgG antibodies in Geneva, Switzerland (SEROCoV-POP): a population-based study. The Lancet. 2020 Aug 1;396(10247):313-9.

Stringhini S, Zaballa ME, Pullen N, Perez-Saez J, de Mestral C, Loizeau AJ, Lamour J, Pennacchio F, Wisniak A, Dumont R, Baysson H. Seroprevalence of anti-SARS-CoV-2 antibodies 6 months into the vaccination campaign in Geneva, Switzerland, 1 June to 7 July 2021. Eurosurveillance. 2021 Oct 28;26(43):2100830.

---

## [Decision Letter · Decision Letter 1]

31 Mar 2022

COVID­-19 pandemic in Saint Petersburg, Russia: combining population­-based serological study and surveillance data

PONE-D-21-27726R1

Dear Dr. Barchuk,

We’re pleased to inform you that your manuscript has been judged scientifically suitable for publication and will be formally accepted for publication once it meets all outstanding technical requirements.

Kind regards,

Siew Ann Cheong, Ph.D.

Academic Editor

PLOS ONE

Additional Editor Comments (optional):

Reviewers' comments:

Reviewer's Responses to Questions

**Comments to the Author**

1. If the authors have adequately addressed your comments raised in a previous round of review and you feel that this manuscript is now acceptable for publication, you may indicate that here to bypass the “Comments to the Author” section, enter your conflict of interest statement in the “Confidential to Editor” section, and submit your "Accept" recommendation.

Reviewer #2: All comments have been addressed

2. Is the manuscript technically sound, and do the data support the conclusions?

Reviewer #2: Yes

3. Has the statistical analysis been performed appropriately and rigorously? 

Reviewer #2: Yes

4. Have the authors made all data underlying the findings in their manuscript fully available?

Reviewer #2: No

5. Is the manuscript presented in an intelligible fashion and written in standard English?

Reviewer #2: Yes

6. Review Comments to the Author

Reviewer #2: I would like to thank the Authors for adressing the comments, I'm satisfied of the responses and recommend publication.

7. PLOS authors have the option to publish the peer review history of their article (what does this mean?). If published, this will include your full peer review and any attached files.

Reviewer #2: No

---

## [Editor Report · Acceptance letter]

6 Jun 2022

PONE-D-21-27726R1 

COVID-19 pandemic in Saint Petersburg, Russia: combining population-based serological study and surveillance data 

Dear Dr. Barchuk:

I'm pleased to inform you that your manuscript has been deemed suitable for publication in PLOS ONE. Congratulations! Your manuscript is now with our production department. 

Kind regards, 

on behalf of

Dr. Siew Ann Cheong 

Academic Editor

PLOS ONE